# Responsiveness to endurance training can be partly explained by the number of favorable single nucleotide polymorphisms an individual possesses

**Henry C. Chung** [1,2]* , **Don R. Keiller**[3] , **Patrick M. Swain**[4] , **Shaun L. Chapman**[2,5] , **Justin D. Roberts** [2] , **Dan A. Gordon**[2]

**1** School of Sport, Rehabilitation and Exercise Sciences, University of Essex, Essex, United Kingdom, **2** Cambridge Centre for Sport & Exercise Sciences, Anglia Ruskin University, Cambridge, United Kingdom, **3** School of Life Sciences, Anglia Ruskin University, Cambridge, United Kingdom, **4** Department of Sport, Exercise, and Rehabilitation, Northumbria University, Newcastle-upon-Tyne, United Kingdom, **5** HQ Army Recruiting and Initial Training Command, United Kingdom Ministry of Defence, Upavon, United Kingdom

☯ These authors contributed equally to this work.
* henry.chung@essex.ac.uk

**Data Availability Statement:** The data underlying the results presented in the study are fully available and presented within the manuscript.

## Abstract

Cardiorespiratory fitness is a key component of health-related fitness. It is a necessary focus of improvement, especially for those that have poor fitness and are classed as untrained. However, much research has shown individuals respond differentially to identical training programs, suggesting the involvement of a genetic component in individual exercise responses. Previous research has focused predominantly on a relatively low number of candidate genes and their overall influence on exercise responsiveness. However, examination of gene-specific alleles may provide a greater level of understanding. Accordingly, this study aimed to investigate the associations between cardiorespiratory fitness and an individual's genotype following a field-based endurance program within a previously untrained population. Participants (age: 29 ± 7 years, height: 175 ± 9 cm, mass: 79 ± 21 kg, body mass index: 26 ± 7 kg/m$^2$) were randomly assigned to either a training (n = 21) or control group (n = 24). The training group completed a periodized running program for 8-weeks (duration: 20-30-minutes per session, intensity: 6–7 Borg Category-Ratio-10 scale rating, frequency: 3 sessions per week). Both groups completed a Cooper 12-minute run test to estimate cardiorespiratory fitness at baseline, mid-study, and post-study. One thousand single nucleotide polymorphisms (SNPs) were assessed *via* saliva sample collections. Cooper run distance showed a significant improvement (0.23 ± 0.17 km [11.51 ± 9.09%], *p* < 0.001, ES = 0.48 [95%CI: 0.16–0.32]), following the 8-week program, whilst controls displayed no significant changes (0.03 ± 0.15 km [1.55 ± 6.98%], *p* = 0.346, ES = 0.08, [95%CI: -0.35–0.95]). A significant portion of the inter-individual variation in Cooper scores could be explained by the number of positive alleles a participant possessed (r = 0.92, R$^2$ = 0.85, *p* < 0.001). These findings demonstrate the relative influence of key allele variants on an individual's responsiveness to endurance training.

**Funding:** There were no funders involved and the authors received no funding for this work.

**Competing interests:** The authors have declared that no competing interests exist.

## Introduction

Cardiorespiratory fitness, often expressed as maximal oxygen uptake ($\dot{V}O_{2max}$), serves as an effective diagnostic and prognostic health marker across a range of populations [1]. Moreover, $\dot{V}O_{2max}$ is positively associated with endurance performance in general, clinical, and athletic populations [2], enabling an individual to increase their time to exhaustion and distance covered when exercising at a fixed or maximal intensity, and improving time-trial performance when exercising at a fixed distance [3]. Even minor increases in cardiorespiratory fitness positively influences general health and functional living, such as walking upstairs without getting out of breath [4]. Accordingly, a better understanding of the factors influencing the degree to which cardiorespiratory fitness can be improved through training, at an individual level, is beneficial for planning future training interventions and monitoring training progress, for both health and exercise performance.

A recent meta-analysis [4] observed that the magnitude of exercise-induced adaptations in the three components of fitness (endurance, strength, and power) were associated with an individual's genotype. Analysis showed that following a period of standardized laboratory exercise training, there were marked inter-individual differences in fitness gains, within groups, in all studies, which could be linked to a genetic component. Specifically, this analysis showed 44%, 72% and 10% of the response variance in aerobic, strength, and power phenotypes, respectively, were explained by genetic influences and subgrouping of genes. Examples of commonly reported genes included but are not limited to ACE, ACTN3, APOE, PGC1a, mTOR and many genes have been shown to influence all aspects of fitness and health, including, energy-pathways, metabolism, cell growth, inflammation, cardiac responses, and protein, hormonal and enzyme interactions and more [4, 5].

These findings highlight the profound effect of an individual's genotype on the degree to which they respond to exercise. However, a major limitation in many of the studies included in the meta-analysis was that improvements in health-related fitness components were often linked to a particular gene, rather than any allelic variation of that gene. Accordingly, there is a relative paucity of research investigating the influence of gene-specific polymorphisms on exercise responses [4]. This is a critical omission, as all individuals share the same genes, but not the same alleles which may, in turn, affect their responsiveness to identical exercise training programs differently [5]. Hence, studies which seek to better understand how an individual's unique genotype may influence their health and fitness responses to exercise training are critical in optimizing future training interventions, recommendations, and pre-screening selection processes [6, 7].

The aim of this study was to engage a previously untrained population of UK adults in an 8-week endurance training program and investigate the influence of allelic variation on cardiorespiratory fitness responses through a proprietary chip containing 1,000 single nucleotide polymorphisms (SNPs). The objectives were to identify specific SNPs associated the increase in cardiorespiratory fitness, rather than the bias "handpicking" of common well explored 'candidate genes' and whether possessing multiple of these 'favorable' SNPs increased an individual's training responsiveness. This study was performed during the onset of the COVID-19 pandemic and, therefore, had to be conducted in a field-based setting as opposed to a standard laboratory environment due to the inability to access the latter for an indefinite amount of time. However, there is a growing body of literature focused on how home-based exercise training can be implemented to combat physical inactivity and to promote improvements in health-related outcomes, including cardiorespiratory fitness [6–8]. Yet, to our knowledge and benefit, there are very limited studies that investigate field-based interventions on measures on cardiorespiratory fitness and their associations with genotypes.

## Materials and methods

### Participants

Sixty-two adults, aged between 20–40 years old, expressed interest, providing written informed consent and were recruited into the study by the end of January 2021. Of these, 45 participants (males: n = 25; females: n = 20) completed the study; 17 withdrew during the study, either due to COVID-19 infection (n = 5) or did not state a reason (n = 12). All participants were British and from the UK Midlands and East Anglia area, confirming that they have not undertaken any endurance training in the previous 8-weeks prior to the start of the study, and were untrained, free from injury, and able-bodied (verified through a pre-health and readiness to exercise questionnaire). Ethical approval was granted by the researchers' institutional ethics board (approval number: FSE/FREP/19/864) (S1 and S2 Files). All participants provided informed consent.

### Study design

A randomized control trial with repeated measures design was employed to investigate the effect of an 8-week field-based endurance running program on cardiorespiratory fitness. This study commenced in January 2021 and was completed in May 2021. Participants were randomly assigned to either an endurance group (EG; n = 21) or control group (CG; n = 24). An *a priori* statistical power analysis determined a sample of n = 18 was required per group ($\alpha$ = 0.05, $\beta$ = 0.80). This calculation was based on studies from a recent meta-analysis with an anticipated change in $\dot{V}O_{2max}$ of 11% post endurance training with similar intervention lengths [4].

A Cooper 12-minute run test [8, 9] was implemented at baseline (week 0), mid-study (end of week 4), and post-study (end of week 8), to assess cardiorespiratory fitness using total distance travelled (km) in the 12-minutes. Participants also self-reported their height and mass on testing days. Upon completion of the training period, a genotyping kit was sent to all participants for the self-collection of DNA using a saliva sample in April 2021.

For all training and assessment sessions, participants were instructed to wear appropriate sportswear, have fasted for at least 3-hours, and remain well-hydrated with water only. All participants recorded their exercise and physical activity habits using an online Excel training diary. This included session duration, estimated exercise intensity (session rating of perceived exertion [sRPE]), and frequency (per week) [10–12]. All data and participant information collected was accessible to the lead researcher (HC). All data was anonymized by the lead researcher before sharing with other authors. This including genetic information that was coded and given unique identities in the first instant by the genetic companies and again was only assessable to lead researcher.

### Cooper 12-minute run test

Participants were instructed to perform a familiarization trial prior to the baseline assessment, to accustom themselves with the protocol. The test required participants to run as far as possible within 12-minutes [8]. Participants were informed of the test protocol through emailed instructions and a telephone discussion was offered if further clarification was required. All participants were instructed to perform each test at the same time of day using the same route on a flat even surface. Using the STRAVA running app (Strava Inc, freemium model), as a Global Positioning System (GPS), participants recorded session distance (kilometers), exercise duration (minutes), location of the run (route), and the time and date it was completed. Session RPE (CR-10) was also recorded 30-minutes after each test for determination of overall

session exertion [13, 14]. Cooper test scores were used as a field-based measure of cardiorespiratory fitness and can be converted to calculate $\dot{V}O_{2max}$. However, it should be noted that this is an indirect, pre-determined, measure in $ml \cdot kg^{-1} \cdot min^{-1}$ that does not include participants mass and therefore, the Cooper run distance completed in km will be reported [8].

## Training intervention

Participants in the EG performed three weekly outdoor runs, increasing in duration from 20 to 30 minutes, over a period of 8-weeks, in accordance with the training schedule shown in Fig 1. For each session, participants aimed for an overall sRPE value of 6–7. Additionally, they were asked to familiarize themselves with the scale and ensure that it was subjectively calibrated to their perceived exertion based on the qualitative descriptors [10].

For training progression, the session duration was increased to enhance the overall training load by 10% each week, whilst sRPE remained constant [6]. Weekly training duration was reduced on the weeks when a Cooper 12-minute run test was completed, ensuring there was still a consistent 10% progression in training load overall (Fig 1). Participants in both groups kept an online training diary that was only accessible to the individual and the lead researcher (HC). Any additional exercises and activities completed over each week of the intervention time-course were recorded. This was regularly monitored for evaluation of participant compliance by the lead researcher (HC). Session RPE was converted into estimated exercise intensity, as an estimated percentage of $\dot{V}O_{2max}$ [15, 16]. For each training session, participant's estimated sRPE ranged from 60–70% $\dot{V}O_{2max}$ [17, 18]. Session training load (sTL), weekly training load (wTL) and total training load (tTL) were calculated for all participants, including any additional session(s) (asTL) and additional weekly training load (awTL) from the information recorded in the training diaries [11, 12].

$$sTL(A.U.) = Running \ duation \ (min) \times session \ intensity \ (\%)$$

$$wTL \ (A.U.) = sTL \ (A.U.) \times number \ of \ sessions \ per \ week$$

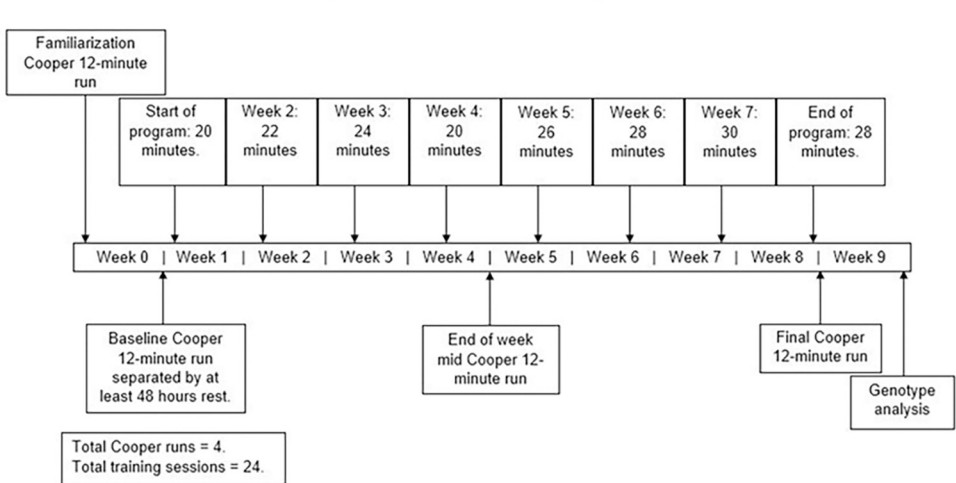

**Fig 1. Training intervention schematic.** Duration of runs per week and Cooper tests. Training progression was increased by 10% of weekly training load by increasing the duration of the run. Intensity was aimed at an estimated 60–70% $\dot{V}O_{2max}$ throughout. Gene test kits and collection of saliva samples were completed post training.

$$tTL\ (A.U.) = wTL\ (A.U.) \times number\ of\ total\ weeks$$

## Genotype analysis

The genetic analysis for this study was performed by Muhdo health Ltd. (Martlesham Heath, Ipswich, England) and Eurofins Global Laboratory (Certification: ISO 17025:2005, ISO 17025:2017). Muhdo supplied the 1,000 SNP custom chip, capable of analysis with a 97–99.9% accuracy level (https://muhdo.com/articles/dna-articles/dna-profiling-your-privacy-and-data/ ). All participants were sent a genotype kit (DNA Health, Muhdo Health Ltd, Ipswich, UK). Participants provided a 2ml passive drool saliva sample in a uniquely coded plastic tube (Gene-FiX™ Saliva DNA/RNA Collection), prefilled with preservative liquid mix (non-toxic stabilization buffer). This allowed the samples to be stored at room temperature and transported, without degradation, for a maximum of 30 days. All samples were sent to Eurofins for genotype analysis. Participants were instructed to not eat, chew gum, brush teeth, use mouth wash, or drink for 3-hours before the sample collection, and to post the samples on the same day.

Phosphate buffered saline was used to prevent any premature cell rupture and cell recovery was accomplished using standard extraction procedures. Recovered cells and assay medium (Cell Lysis Solution Infinium, GoldenGate) were loaded into chip wells for Illumina multiplex sequencing. The custom chips (Illumina® Infimum HumanOmni BeadChip) with bound DNA sequences, containing target SNPs, were scanned using a Microarray Scanner (iScan, Illumina, San Diego, CA, USA). Fluorescence information was collated using Illumina's standard software (Illumina's GenomeStudio® software). The iScan Control Software automatically normalized the intensity of the fluorescence data to remove technical variation and generated genotype calls [19]. Three probes were used, in both forwards and reverse orientations and repeated three times to ensure accurate determination of alleles. Nucleotides (Adenine, Thymine, Guanine and Cytosine) were listed in the forward-forward orientation. If this information was not consistent, in all calls, the allele set was not included. All internally consistent genetic data was exported to an Excel spreadsheet (Microsoft Corp, Washington, USA), listing the SNP's by rs number.

## Statistical analysis

All data are reported as mean ± standard deviation (SD) and data analysis was performed using the statistical package JASP 0.16.2.0. (JASP team 2017) unless otherwise specified. All variables were assessed for parametric assumptions using the Shapiro-Wilk test of normality and Levene's test for homogeneity of variance. Statistical significance was set at $p \leq 0.05$. Independent sample t-tests were used to assess the effect of the training intervention (EG vs. CG) on outcome measurements at week 4 and 8. Further, a one samples t-test was used to compare the within group changes in Cooper scores between participants. A repeated measures analysis of variance (ANOVA) was used to assess differences in outcomes within the EG and CG between baseline, week 4, and week 8 results. Equivalent non-parametric tests were used where data failed to meet parametric assumptions.

Standardized mean differences (SMD) were calculated, using Cohen's *d* effect size (ES), with 95% Confidence intervals (CI). The ES was defined by the following thresholds: small (ES = 0.20–0.49), medium (ES = 0.50–0.79), and large (ES $\geq$ 0.80) [20]. A sizeable percentage of the SNPs (285; 28.5%) showed no allelic variation, between participants and were manually removed from the dataset, prior to statistical analysis. Potential exercise associated SNPs were identified using Pearson's chi-square test ($\chi^2$). $X^2$ analysis was used to screen the 1,000 SNP database for significant associations. A total of 25 SNPs were identified as having alleles which

were significantly ($p \leq 0.05$) associated with a greater than average Cooper run improvement. Another 23 SNPs were in the range $p = 0.051–0.075$. Since the $\chi^2$ analysis provided no information about the magnitudes of each participant's training response, ANOVA was employed to assess the effect of individual alleles on the training response magnitude.

SNPs associated with improved cardiorespiratory fitness identified by the ANOVA were summed by allele for each participant according to the following procedure. Homozygosity for alleles associated with improved exercise performance scored "2", whilst heterozygosity scored "1" and homozygosity for no positive alleles scored "0". The resultant allelic scores were summed and subjected to linear regression analysis against the percentage improvements in Cooper test scores. Allelic frequencies associated with improved Cooper run scores, were compared with expected European population frequencies (https://www.nlm.nih.gov/; https://selfdecode.com/; https://www.snpedia.com/), or where this information was not available, calculated from published reference and alternate allele proportions, using the Hardy-Weinberg formula. With relatively low sample sizes and in some cases, zero values for some allelic combinations, frequency comparisons were made using Fisher's exact test [21–24].

## Results

### Participant characteristics

Forty-five participants completed the 8-week study intervention (S1 Data). Participant characteristics were as follows: EG (n = 21), age: 31 ± 9 years, height: 178.8 ± 8.5 cm, mass: 82.1 ± 17.1 kg, and BMI 25.5 ± 4.0 kg/m$^2$ and CG (n = 24), age: 28 ± 5 years, height: 171.8 ± 8.4 cm, mass: 73.1 ± 15.4 kg, and BMI 24.7 ± 4.2 kg/m$^2$. There were no significant differences in any baseline characteristics between the EG and CG. In total, the EG had an average sTL = 2,421 ± 861 A.U., wTL = 6,321 ± 1,251 A.U., and tTL = 50,811 ± 10,347 A.U. Training diaries of the CG revealed that some participants engaged in exercise during the study with an average sTL = 1,461 ± 1,026 A.U., wTL of 3,285 ± 2,599 A.U., and tTL of 23,513 ± 20,471 A.U. However, the EG had significantly larger training loads compared to the CG by 66% (U = 75.00, $p < 0.001$, ES = 1.01 [95%CI: 0.39–1.63]), 92% (U = 50.00, $p < 0.001$, ES = 1.46 [95%CI: 0.80–2.11]), and 116% (U = 43.00, $p < 0.001$, ES = 1.65 [95%CI: 0.98–2.33]), respectively.

### Cooper 12-minute distance

Both groups were normally distributed for baseline Cooper scores, (EG = D (21), 0.972, $p = 0.779$; CG = D (24), 0.200, $p = 0.724$) and were homogenous ($F$ (1,43) = 0.941, $p = 0.337$). Here, ANOVA revealed no significant differences between baseline Cooper scores between the EG (2.22 ± 0.48km) and CG (2.06 ± 0.37km) ($F$ (1, 43) = 1.623, $p = 0.209$).

The EG significantly increased their running distance at week 4 (0.14 ± 0.19 km, t (20) = -3.392, $p = 0.003$, ES = 0.30 [95%CI: 0.05–0.23]) and week 8 (0.23 ± 0.17 km [11.51 ± 9.09%]) (t (20) = 6.271, $p < 0.001$, ES = 0.48 [[95%CI: 0.16–0.32]). The CG showed no significant changes relative to baseline scores at week 4 (0.04 ± 0.13 km [$p = 0.127$, ES = 0.10; 95%CI: -0.013–0.99]) or week 8 (0.03 ± 0.15 km [1.55 ± 6.98%], $p = 0.346$, ES = 0.08, [95%CI: -0.35–0.95]). Fig 2. Illustrates changes in Cooper run score across the study period between the EG and CG.

Between group analysis showed that Cooper distance was significantly greater in the EG than the CG at week 8 ($p = 0.048$; ES = 0.62 [95%CI: 0.01–1.21]) and week 8 ($p = 0.007$, ES = 0.85 [95%CI: 0.24–1.46]). Interestingly, regression analysis showed a significant association between baseline Cooper run performance and the percentage improvement after 8-weeks, with higher percentage improvement being associated with lower baseline Cooper

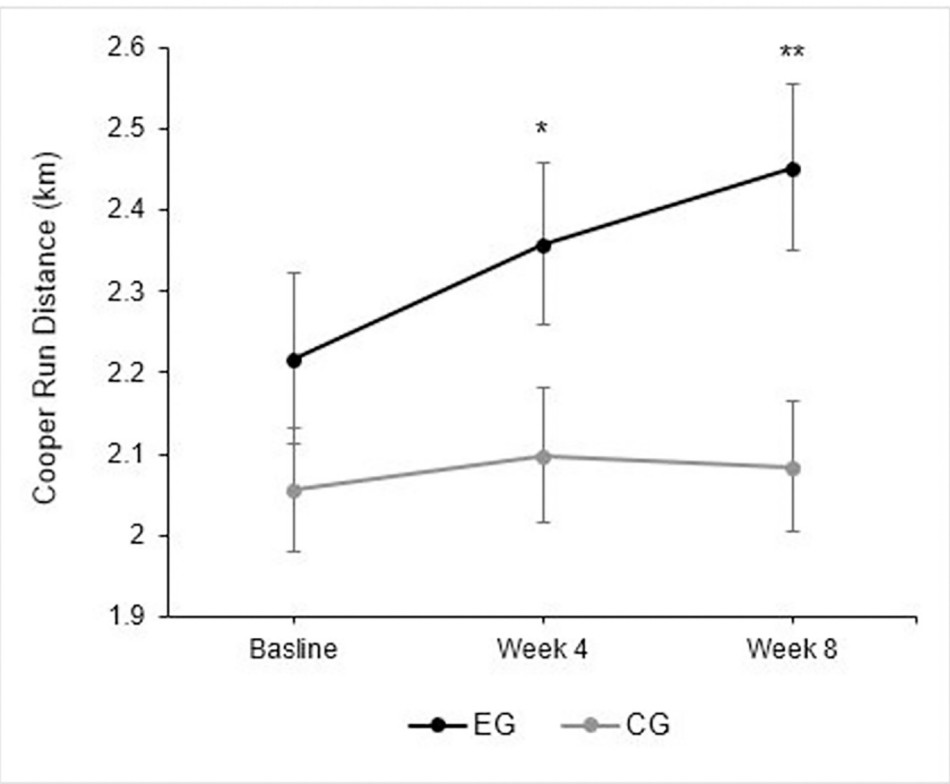

**Fig 2. Coper 12-minute responses in exercise and control group.** Cooper run response across the 8-week intervention with standard error (SE) bars. Where Cooper Run tests were performed in week 0, 4 and 8; * = significance level at $p < 0.01$; ** = significance level at $p < 0.001$.

run scores (r = 0.71, $R^2$ = 0.50, $p < 0.001$), indicating the effect of training status. In the context of this study, the one samples t-test showed that there were marked differences between participants in the EG ($p < 0.001$ [95%CI: 10.42–23.12]), demonstrating that individuals responded differently to the same endurance-based training program (Fig 3). No such difference was identified in the CG ($p = 0.337$ [95%CI: -2.35–6.54]).

## Genotypes

A total 19 SNPs were found to be significantly associated to the greater than average improvement in Cooper run performance ($p \leq 0.05$), shown in Table 1. Moreover, there was a significant positive correlation between the summed allele scores (i.e., the number of 'favorable' SNPs a participant possessed) and the change in Cooper run scores post-training (r = 0.92, $R^2$ = 0.85, $p < 0.001$) (Fig 4). The participant who showed a decrease in performance after the endurance intervention had no positive alleles for the six most significant SNPs (BDNF; BRINP3; KL [rs7997728]; KL [rs9527021]; MAOA; MICB). Table 1 summarizes the allele subgroups that are associated with improvements in cardiorespiratory fitness. All SNPs identified by ANOVA demonstrated homogeneity of variance and were included in the final analysis. Fisher's Exact test found no significant differences in the allele frequencies observed in this study and those expected within the European population with the exception of rs3829116 (NOTCH1) that was consequently excluded from analyses.

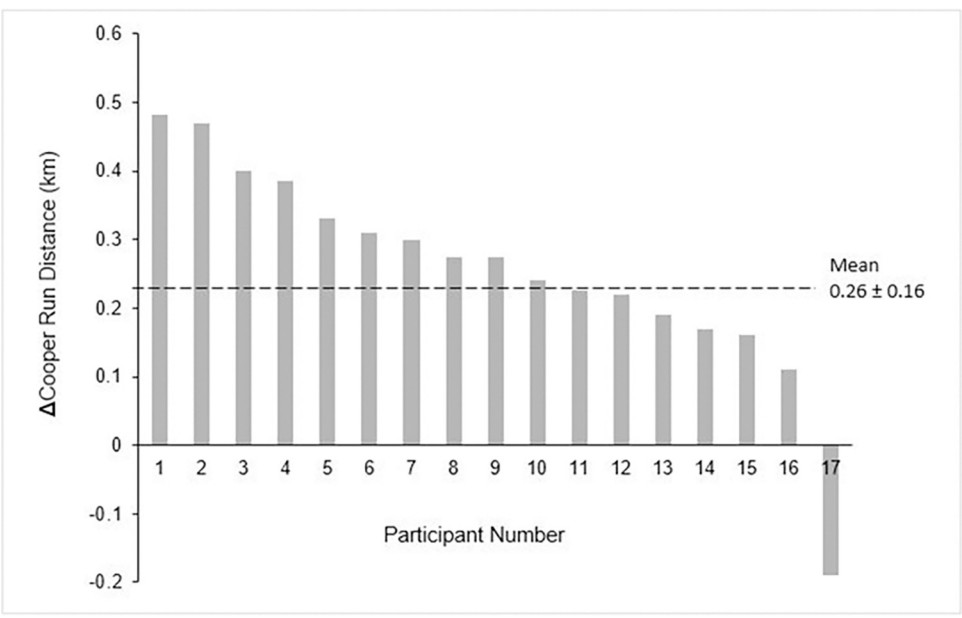

**Fig 3. Waterfall plot.** The change in running distance response in the exercise group, post 8-weeks training. The black dashed line represents the average improvement post training in km. Each bar represents a single participant. Of the 21 participants, participants 6, 8, 12, and 14 were removed from this analysis because their genetic data was not internally consistent.

## Discussion

The aim of this study was to investigate genetic associations with cardiorespiratory fitness following an endurance-based program within a previously untrained population. Participants within the EG significantly increased Copper 12-minute running distance by an average of 11.5% whereas the control group displayed no significant changes. Eighteen SNPs were positively associated with an above average increase in running distance and the number of these 'favorable' SNPs participants possessed displayed a significant linear correlation with the relative change in Cooper scores post-training (r = 0.92, $R^2$ = 0.85; $p < 0.001$). These findings, therefore, demonstrate that inter-individual variability in the responsiveness to an endurance training program can be largely explained by the number of 'favorable' SNPs an individual possesses. Such findings concur with previous literature statements and laboratory-based studies, which have reported associations with $\dot{V}O_{2max}$ scores attributable to genetic factors [4, 5, 25, 26]. However, to our knowledge this is the first study to have established a relationship between cardiorespiratory fitness and allele-specific genotypes using field-based measures and training.

Of these 18 SNPs, nine were found in regions of the genome that do not code for proteins (Table 1). Such non-coding DNA (ncDNA) accounts for the majority of the human genome and includes sequences that are transcribed into a variety of RNA molecules, such as ribosomal RNAs, microRNAs, long non-coding RNAs, plus un-transcribed sequences that have regulatory functions, including gene promoters and enhancers [27]. These are likely to be cis-regulatory elements (CREs). CREs are DNA sequences containing binding sites for transcription factors (TFs) and/or other regulatory molecules that are required to activate, modulate and sustain transcription [28]. As well as these ncDNA SNPs, five SNPs were located within the introns of coding genes (Table 1). Introns, like ncDNA, often have regulatory functions, such as controlling alternative splicing, enhancing gene expression, controlling mRNA transport, and affecting mRNA nonsense-mediated decay, all of which are highly relevant in the context

**Table 1. List of candidate genes.** Table shows SNPs, associated with Cooper test improvement, listed in order of $\chi^2$ significance. Also listed are the respective genes, SNP type. Fisher Exact test statistic, ANOVA significance and the allele associated with improved exercise performance.

| SNP | Gene | Alleles | Type | Position | Fisher Exact | p ($\chi^2$) | p (ANOVA) | Positive Allele | Coding? |
|---|---|---|---|---|---|---|---|---|---|
| Rs41293864 | MICB | C>T | Intron Var | chr6:31466217 | 0.719 | 0.001 | 0.04 | T | No |
| rs11800795 | BRINP3 | A>C | Downstream SNV | chr1:189974258 | 0.13 | 0.002 | 0.008 | C | No |
| rs909525 | MAOA | G>A | intron Var | chrX:43693955 | 0.386 | 0.004 | 0.001 | G | No |
| rs7997728 | KL | T>G | Intron Var | chr13:33054420 | 0.719 | 0.006 | 0.003 | G | No |
| rs925946 | BDNF | T>G | Downstream SNV | chr11:27645655 | 0.141 | 0.006 | 0.001 | T | No |
| rs9527021 | KL | A>G | Intron Var | chr13:33052885 | 0.844 | 0.006 | 0.003 | G | No |
| rs6473227 | FABP5[a] | C>A | Downstream SNV | chr8:81285892 | 0.503 | 0.008 | 0.005 | A | No |
| rs1815739 | ACTN3 | C>T | Stop codon gain | chr11:66560624 | 0.663 | 0.022 | 0.043 | C | Yes |
| rs12926089 | CLCN7 | C>T | Missense cod var. | chr16:1452856 | 0.699 | 0.023 | 0.005 | T | Yes |
| rs16998073 | FGF5 | A>T | Upstream SNV | chr4:80263187 | 0.317 | 0.024 | 0.033 | A | No |
| rs7323281 | KL | A>G | Intron Var | chr13:33053623 | 0.866 | 0.024 | 0.009 | A | No |
| rs3088440 | CDKN2A | G>A | 3'UTR Var | chr9:21968160 | 0.699 | 0.029 | 0.042 | A | No |
| rs6473223 | FABP5[a] | T>C | Upstream SNV | chr8:81268155 | 0.561 | 0.034 | 0.05 | C | No |
| rs3829116 | NOTCH1[b] | C>G | Intron Var | chr9:136507125 | 0.004 | 0.034 | 0.025 | G | No |
| rs17782313 | MC4R | T>C | Downstream SNV | chr18:60183864 | 0.319 | 0.036 | 0.049 | C | No |
| rs6563 | NOTCH1 | A>G | 3' UTR Var | chr9:136494732 | 0.912 | 0.037 | 0.033 | A | No |
| rs2229974 | NOTCH1 | C>T | Coding-synon/Asp>Asp | chr9:136497184 | 0.303 | 0.049 | 0.027 | T | Yes |
| rs2837960 | BACE2 | T>G | Upstream SNV | chr21:41139991 | 0.844 | 0.056 | 0.038 | T | No |
| rs2108622 | CYP4F2 | C>T | Missense cod var. | chr19:15879621 | 0.492 | 0.059 | 0.005 | C | Yes |

Notes: Hardy-Weinburg equilibrium.(HWE) frequencies were found, or calculated, where necessary using https://www.nlm.nih.gov/; and https://opensnp.org/

[a] Gene containing SNP, or closest protein-coding gene.

[b] SNP excluded from further analysis because significantly different from HWE. SNV = single nucleotide variation; Intron Var = Intron Variant; 3' UTR Var = 3_prime Untranslated Region Variant; Positive allele = allele associated with increased exercise performance.

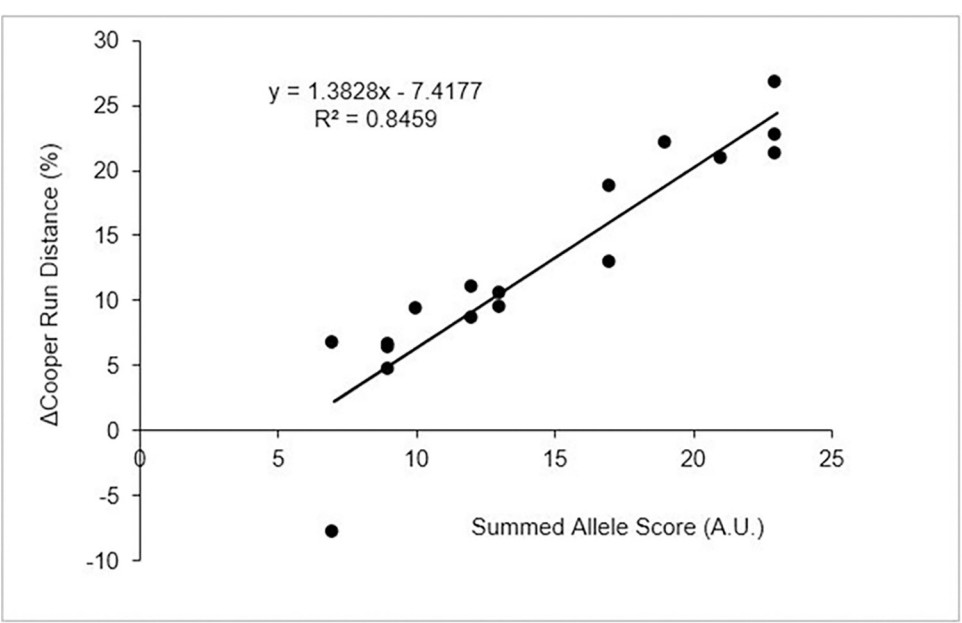

**Fig 4. Allele and Cooper run regression.** Regression plot illustrating the number of positive alleles a participant possessed is significantly associated with the relative change in Cooper 12-minute run test score.

of this study [29]. The remaining four SNPs were missense single nucleotide variation (SNVs). Although not all missense mutations cause a change in the overall structure and function of the protein, the resulting protein may not be as active, show altered responses to regulatory factors, may be less stable, or may fail to localize in its proper intracellular position [30].

At this point we note that one of the criticisms that can be levelled against this study is that it used a relatively small number of participants and many statistical analyses, hence the probability of identifying false positives (type I statistical error) is high. Under such circumstances the Bonferroni correction [31] is often considered necessary. However, four observations from this study suggest that the exercise associated alleles have been identified correctly, rather than by chance, hence the Bonferroni correction is inappropriate under such circumstances.

1. Within the 18 SNPs identified, three genes (KF three-times; FABP3, and NOTCH1, twice) are disproportionately represented. It is highly unlikely that a set of randomly selected false positives would repeatedly identify the same gene. The probability of identifying the KL gene three times, independently, at random, is $1.25 \times 10^{-4}$ ($0.05^3$), meaning that the expected probability for this gene to be randomly identified from a dataset of 715 SNPs is 0.089%.

2. Examination of the chromosomal positions of the KL gene (https://opensnp.org/snps/) showed that rs7197728 was located between exon 2 and 3, whilst rs9527021 and rs7313281 were both upstream of exon 2 but separated by 738 bases. Similarly, the two FABP5 and NOTCH1 SNPs were also clearly different (Table 1). This, again, indicates that an independent selection of SNPs had been made.

3. If the SNPs identified had a significant relationship, purely by chance, with improved Cooper test scores, it is highly unlikely that their effects would be additive, as shown in Fig 4.

4. Finally, the analysis identified ACTN3, a well-known fitness associated gene that was also highlighted within the recent meta-analysis [4]. Again, the probably of this happening by chance is extremely low. The SNP, rs1815739 in the ACTN3 gene, encodes a premature stop codon in a muscle protein called alpha-actinin-3. The polymorphism alters position 577 of the alpha-actinin-3 protein. In publications the (CC) genotype is often called RR, whereas the (TT) genotype is often called XX [32].

An interesting common factor noted in this study was the association of the minor alleles of the SNPS rs41293864, rs11800795, rs7997728, rs9527021, rs12926089 and rs3088440, which tend to promote inflammation, with increased exercise performance (Table 1). Seven (MICB, [33], BRINP3 [34], KL[35, 36] CLCN7 [37, 38], CDKN2A, [39] BACE2 [40], CYP4F2 [41, 42]) of the 14 genes identified have previously being linked with increased levels of oxidative stress and inflammation [36, 43], processes that can damage many body systems, including the lungs [44], heart [34] and blood vessels [45]. Here the possibility is that low level inflammation, associated with the possession of such alleles, may adversely affect baseline fitness. In this connection it is noteworthy that physical exercise can help reduce such inflammation [40], thus accounting for the improved exercise performance observed in this study. Another possible explanation of this link with inflammation associated alleles is antagonistic pleiotropy, which hypothesizes that genes that enhance fitness early in life may diminish it, in later life [46]. The remaining 7 genes, associated with improved exercise performance, do not appear to share any unifying common factors and are therefore discussed separately.

## MAOA gene (rs909525)

Located on the X-chromosome the rs909525 polymorphism is the best proxy for the number of repeats of the monoamine oxidase A (MAOA) gene, also known as the "warrior" gene,

because it is linked to aggression [47, 48]. Those that possess 4–5 repeat A-alleles are classed as "non-warrior" and those with 3-repeat G-alleles are likely to be "warriors". In "warrior" rats a decrease in MAOA levels and lowered monoamine oxidase activity, markedly increased wheel-running activity [49]. This is reflected in this study with those that possess the warrior allele showing improved exercise performance.

### BDNF gene (rs925946)

Brain Derived Neurotropic factor (BDNF) helps regulate the management of eating, drinking, and body weight [50]. Rs4923461 is a downstream SNV and most likely serves to regulate BDNF transcription [27]. BDNF levels have previously been reported to be inversely associated with fasting plasma glucose among type II diabetes patients and associated with the severity of insulin resistance [51]. This suggests that BDNF regulates blood glucose homeostasis and insulin sensitivity. Using quantitative trait analysis, one study [51] found that the G-allele was the risk-allele associated with elevated fasting plasma glucose levels. In this study the non-risk allele (T) was associated with improved running distance.

### FABP5 gene (rs6473227; rs6473223)

The nearest gene to these intergenic variants is FABP5 (Table 1). As lipid chaperones, FABP5 has an important role in regulating the transport of intracellular fatty acids [52]. Furthermore, FABP5 modulates the concentrations of fatty acids and their CoA, and carnitine esters, which have a key role in oxidative energy production [53]. There is also evidence [54] FABP5 participates in the uptake of circulating fatty acids into cardiac and skeletal myocytes to maintain ATP levels. Here acetyl-CoA availability in the mitochondrial matrix adjusts fatty acid oxidation to exercise intensity and duration [55]. Accordingly, changes in the levels of FABP5, possibly controlled by intergenic sequences [27], are likely to impact on fatty acid metabolism, a key component of aerobic exercise energy production and adaptation.

### ACTN3 gene (rs1815739)

Interestingly, the ACTN3 rs1815739 CC (RR) genotype has previously been associated with power and sprinting performance, rather than endurance, whilst the TT (XX) genotype is typically associated with aerobic fitness [56, 57]. Our findings, however, support more recent findings [58, 59] that the CC genotype is associated in endurance improvements. A recent study [60] showed that exercise stimulates the production of N-lactoyl-phenylalanine (Lac-Phe), a blood-borne signaling metabolite. The more lactate produced, the more substrate for Lac-Phe production. Elevations in plasma Lac-Phe levels, following physical activity have been found in mice, racehorses, and humans, establishing this metabolite as a key molecular effector associated with physical activity. Data from a human exercise cohort [60] showed that sprint exercise induced the most dramatic increase in plasma Lac-Phe, followed by resistance training and then endurance training. Muscle cells with the C-allele, rely more on anaerobic respiration, which will increase lactate production, resulting in an increase in N-lactoyl-phenylalanine synthesis. This in turn, will promote positive exercise responses [60], as was found in this study.

### FGF5 gene (rs16998073)

Fibroblast growth factor 5 (FGF5) possess broad mitogenic and cell survival activities, and are involved in a variety of biological processes, including embryonic development, cell growth, morphogenesis, tissue repair, tumor growth and invasion [61–63]. Notably, for the rs16998073 variant, the hypertension-related risk-allele (T) has been shown to significantly reduce exercise

performance, [62]. Such observations agree with the finding of this study, where the non-risk allele (A) was associated with improved exercise performance.

### MC4R gene (rs17782313)

Melanocortin-4-receptor is one of the key receptors involved in central regulation of energy homeostasis [64]. MC4R gene has been established as the second important locus associated with BMI [65]. Rs17782313, is a downstream SNV which may influence gene function by modulating MC4R expression [27]. Although the rs17782313 C-allele is considered a risk factor for obesity with high energy intake [66], a recent meta-analysis [67] determined that the heterozygote does not have a significant connection with obesity. The results in this study support these findings, as the cohort did not have any significant changes in bodyweight and those that improved exercise performance, were heterozygotic.

### NOTCH1 gene (rs6563; rs2229974)

The Notch 1 gene codes for a receptor protein, which helps determine cell specialization [68]. Mutations in the NOTCH1 gene, such as rs6563 (G allele), can impair normal heart development before birth. This likely affects the heart's ability to circulate blood and therefore, the ability to exercise for prolonged periods or at high intensities, which will impact performance adaptations [69]. In agreement, this study showed that the AA genotype was significantly associated with improvied cardiorespiratory fitness, compared to those that had the risk allele. Since rs6563 is a 3' UTRs SNV, it could impact on mRNA-based processes, such as mRNA localization, mRNA stability, and translation [70]. Theoretically the rs2229974 (coding-synon/Asp>Asp), should have no effect, but may show linkage disequilibrium to other polymorphisms. In this connection, a study [71] investigating the role of NOTCH1 mutations in Aortic Stenosis (AS), found a highly significant association of rs2229974 with AS, which will likely have an impact on an individual's exercise performance ability.

### Additional remarks

The lack of consistency in precisely defining an allele's role in response to exercise is not surprising. Previous research has typically been based on observational studies on specific populations [72, 73] or focusing mainly on the frequencies and distributions of the alleles [74, 75]. In contrast, this study focused on identifying particular alleles, which were associated with improved cardiorespiratory fitness, in untrained participants, subjected to an endurance training program. In this context, candidate genes may be a better potential indicator of overall performance and health, rather than predicting specific phenotypic outcomes alone [75–77]. Furthermore, studies have shown that multiple genes and their alleles can participate in many pathways, contributing to the same functions, meaning the apparent influence of individual SNPs may vary from person to person [78]. In fact, previous Genome-wide association studies (GWAS) established that complex multifactorial traits, such as cardiorespiratory fitness, are influenced by polygenic systems, characterized by relatively common alleles, each with relatively small effect sizes [78]. As was the case in this study, many of these allelic variants are not located in coding regions and a large fraction of them reside in the noncoding regulatory genome. Additionally, functionally equivalent isoenzymes can be subject to differential control and expression in response to endurance training [79, 80]. Hence the situation is highly complex, and no common panel of genetic variants have yet been identified, at a genome wide significance level, that can predict how well someone will respond to exercise training [81]. Hence further research is required to elucidate the links between individual genetic polymorphisms in determining exercise and performance.

## Limitations

The inability to perform laboratory-based research due to COVID-19 restrictions did impede our ability to rigidly standardize, control, and supervise the training and testing. Yet, the Cooper 12-minute run test overcame several practical constraints to estimating cardiorespiratory fitness, within a field-based setting, during a period of high COVID-19 restrictions. Therefore, laboratory-based exercise protocols are warranted.

We recognize that the control group should have refrained from exercise and adhered to their regular routines. However, due to participants being confined to their residences, uptake of self-directed physical activity was not uncommon during the UK national lockdown. Though, to this studies benefit, this additional uptake in physical activity did not result in any significant improvements in Cooper run test scores in the control group.

Finally, a larger sample size for the allele subgroups is necessary and is a limitation of this study. In particular, low sample size meant that allele could not be identified i.e. for 28% of the SNPs alleles with low frequencies were not represented within our cohort and had to be removed, prior to analysis. Consequently, such SNPs should not be disregarded as insignificant as some rare variants could be identified with larger sample sizes. Therefore, greater sample sizes are necessary for genetic studies.

## Conclusion

The study demonstrates that an 8-week field-based endurance training program elicits significant improvements in cardiorespiratory fitness within a previously untrained population, relative to a non-exercise control group. Significant inter-individual variability in endurance performance gains were observed following the training program. Eighteen SNPs were significantly associated with a greater than average Cooper run test score and a significant portion of the variation in endurance performance was explained by how many of these SNPs an individual possessed ($r = 0.92$, $R^2 = 0.85$, $p < 0.001$). The use of allele-specific analysis and identifying genetic polymorphisms provides a deeper understanding to the influence of genetics on exercise phenotypes, when compared to many previous exercise studies that have examined associations with particular genes only. These findings indicate there are allele variants which produce superior improvements in cardiorespiratory fitness than their counterparts and that their effects are additive. This knowledge helps improve our understanding of the genetic influences underlying individuals' responsiveness to endurance training.

## Supporting information

**S1 File. Original ethics approval letter.** Pre COVID-19 application and approval.
(PDF)

**S2 File. Amended ethics approval letter.** The protocol to complete the project was edited in response to COVID-19.
(PDF)

**S1 Data. Participant data.** Individual data collected for each participant.
(PDF)

## Acknowledgments

We thank all the participants who were involved in this study. We would especially like to thank Mr. Christopher Collins and Mr. Richard Layton at MUHDO Health Ltd and Eurofins Laboratory for organizing the genetic analysis for this study and offering logistical support

during the COVID-19 pandemic. This research did not receive any specific grant or funding from any agencies in the public, commercial, or for-profit sectors. All relevant data is publicly available and within the contents of this manuscript. Additional data can be obtained upon reasonable request to the corresponding author.

## Author Contributions

**Conceptualization:** Henry C. Chung, Justin D. Roberts, Dan A. Gordon.

**Data curation:** Henry C. Chung, Dan A. Gordon.

**Formal analysis:** Henry C. Chung, Don R. Keiller, Patrick M. Swain.

**Investigation:** Henry C. Chung.

**Methodology:** Henry C. Chung, Don R. Keiller, Dan A. Gordon.

**Project administration:** Henry C. Chung, Justin D. Roberts, Dan A. Gordon.

**Resources:** Henry C. Chung, Don R. Keiller, Patrick M. Swain, Shaun L. Chapman, Justin D. Roberts, Dan A. Gordon.

**Software:** Don R. Keiller.

**Supervision:** Don R. Keiller, Justin D. Roberts, Dan A. Gordon.

**Validation:** Henry C. Chung, Don R. Keiller, Patrick M. Swain, Shaun L. Chapman, Dan A. Gordon.

**Visualization:** Henry C. Chung, Justin D. Roberts, Dan A. Gordon.

**Writing – original draft:** Henry C. Chung, Don R. Keiller, Patrick M. Swain, Shaun L. Chapman, Justin D. Roberts.

**Writing – review & editing:** Henry C. Chung, Don R. Keiller, Patrick M. Swain, Shaun L. Chapman, Justin D. Roberts, Dan A. Gordon.

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
