## [Decision Letter · Decision Letter 0]

12 Apr 2023

PONE-D-22-31088Responsiveness to endurance training can be partly explained by the number of favorable single nucleotide polymorphisms an individual possessesPLOS ONE

Dear Dr. Chung,

Thank you for submitting your manuscript to PLOS ONE. After careful consideration, we feel that it has merit but does not fully meet PLOS ONE’s publication criteria as it currently stands. Therefore, we invite you to submit a revised version of the manuscript that addresses the points raised during the review process.

We look forward to receiving your revised manuscript.

Kind regards,

Chunyu Liu, PhD

Academic Editor

PLOS ONE

Journal Requirements:

Additional Editor Comments:

Dear authors,

Thank you for submitting the manuscript to PLOS ONE. We could not accept your manuscript at the current stage and would like to invite you to resubmit your revision by addressing the reviewer's comments.

Reviewers' comments:

Reviewer's Responses to Questions

**Comments to the Author**

1. Is the manuscript technically sound, and do the data support the conclusions?

Reviewer #1: Yes

2. Has the statistical analysis been performed appropriately and rigorously? 

Reviewer #1: Yes

3. Have the authors made all data underlying the findings in their manuscript fully available?

Reviewer #1: No

4. Is the manuscript presented in an intelligible fashion and written in standard English?

Reviewer #1: Yes

5. Review Comments to the Author

Reviewer #1: Comments:

The manuscript is interesting but requires modifications before publishing.

1. Novelty of the research should be emphasized more in the introduction section.

2. More explanation regarding the EG participant who showed decrease in performance after the endurance intervention would be of interest.

3. A characteristic table of the participants would be helpful.

4. How did you improve and ensure the reproducibility of the study?

5. Any implications on future studies based on your conclusion?

6. Grammar-related errors and minor formatting-related errors and need to be rectified.

6. PLOS authors have the option to publish the peer review history of their article (what does this mean?). If published, this will include your full peer review and any attached files.

Reviewer #1: No

---

## [Author Response · Author response to Decision Letter 0]

14 Apr 2023

Thanks again for reviewing the manuscript and sending us the comments. This manuscript was initially fully submitted to PLOS Genetics, however has been requested for a transfer to PLOS One. I have now included the manuscript with track changes, manuscript with accepted changes, rebuttal letter, cover letter with edited statement confirming no funding, and edited supporting information. 

I believe all is in accordance now and that we have addressed all comments that were provided. Please do not hesitate to contact me if you require anything else. 

Thank you

---

## [Decision Letter · Decision Letter 1]

10 Jul 2023

Responsiveness to endurance training can be partly explained by the number of favorable single nucleotide polymorphisms an individual possesses

PONE-D-22-31088R1

Dear Dr. Chung,

We’re pleased to inform you that your manuscript has been judged scientifically suitable for publication and will be formally accepted for publication once it meets all outstanding technical requirements.

Kind regards,

Alvaro Galli

Academic Editor

PLOS ONE

Additional Editor Comments (optional):

Reviewers' comments:

Reviewer's Responses to Questions

**Comments to the Author**

1. If the authors have adequately addressed your comments raised in a previous round of review and you feel that this manuscript is now acceptable for publication, you may indicate that here to bypass the “Comments to the Author” section, enter your conflict of interest statement in the “Confidential to Editor” section, and submit your "Accept" recommendation.

Reviewer #1: All comments have been addressed

2. Is the manuscript technically sound, and do the data support the conclusions?

Reviewer #1: Yes

3. Has the statistical analysis been performed appropriately and rigorously? 

Reviewer #1: Yes

4. Have the authors made all data underlying the findings in their manuscript fully available?

Reviewer #1: No

5. Is the manuscript presented in an intelligible fashion and written in standard English?

Reviewer #1: Yes

6. Review Comments to the Author

Reviewer #1: (No Response)

7. PLOS authors have the option to publish the peer review history of their article (what does this mean?). If published, this will include your full peer review and any attached files.

Reviewer #1: No

---

## [Editor Report · Acceptance letter]

12 Jul 2023

PONE-D-22-31088R1 

Responsiveness to endurance training can be partly explained by the number of favorable single nucleotide polymorphisms an individual possesses 

Dear Dr. Chung:

I'm pleased to inform you that your manuscript has been deemed suitable for publication in PLOS ONE. Congratulations! Your manuscript is now with our production department. 

Kind regards, 

on behalf of

Dr. Alvaro Galli 

Academic Editor

PLOS ONE